palaeontology/evolution

diversification rates, Dinosauria, GLMMs, phylogeny, Bayesian, K-Pg boundary

**Author for correspondence:**
Joseph A. Bonsor
e-mail: joseph.bonsor@gmail.com

# Dinosaur diversification rates were not in decline prior to the K-Pg boundary

Joseph A. Bonsor[1,3], Paul M. Barrett[1],
Thomas J. Raven[1,4] and Natalie Cooper[2]

[1]Department of Earth Sciences, and [2]Department of Life Sciences, Natural History Museum, Cromwell Road, London SW7 5BD, UK
[3]Department of Biology and Biochemistry, University of Bath, Claverton Down, Bath BA2 7AY, UK
[4]School of Environment and Technology, University of Brighton, Lewes Road, Brighton BN2 4GA, UK

JAB, 0000-0002-8829-2778; PMB, 0000-0003-0412-3000; TJR, 0000-0002-4349-5635; NC, 0000-0003-4919-8655

Determining the tempo and mode of non-avian dinosaur extinction is one of the most contentious issues in palaeobiology. Extensive disagreements remain over whether their extinction was catastrophic and geologically instantaneous or the culmination of long-term evolutionary trends. These conflicts have arisen due to numerous hierarchical sampling biases in the fossil record and differences in analytical methodology, with some studies identifying long-term declines in dinosaur richness prior to the Cretaceous–Palaeogene (K-Pg) boundary and others proposing continued diversification. Here, we use Bayesian phylogenetic generalized linear mixed models to assess the fit of 12 dinosaur phylogenies to three speciation models (null, slowdown to asymptote, downturn). We do not find strong support for the downturn model in our analyses, which suggests that dinosaur speciation rates were not in terminal decline prior to the K-Pg boundary and that the clade was still capable of generating new taxa. Nevertheless, we advocate caution in interpreting the results of such models, as they may not accurately reflect the complexities of the underlying data. Indeed, current phylogenetic methods may not provide the best test for hypotheses of dinosaur extinction; the collection of more dinosaur occurrence data will be essential to test these ideas further.

## 1. Introduction

Dinosaurs were the most conspicuous members of late Mesozoic terrestrial ecosystems until the impact of an approximately 10 km wide meteorite caused the extinction of all non-avian dinosaur taxa 66 million years ago (Ma) [1–4]. The effects of this event were

**Figure 1.** The three models used in this study. (*a*) The null model, where node count is a linear function of time elapsed; (*b*) the slowdown to asymptote model, where node count is a function of $\sqrt{\text{time elapsed}}$ and (*c*) the downturn model, where node count is a function of time elapsed and its quadratic term.

globally devastating and were caused by an ejecta blanket of dust and debris that encircled the Earth and cooled global temperatures by up to 10°C for decades, alongside the more immediate effects of wildfires and tsunamis in areas proximal to the impact [2,5–7].

Determining trends in Mesozoic dinosaur species-richness immediately prior to the K-Pg boundary is a contentious issue and has been the focus of numerous previous studies (e.g. [3,8–10]). Many different hypotheses have been proposed, but there are currently three schools of thought. The first, reviewed extensively by Brusatte *et al.* [3], suggests that there is little evidence for a global downturn in dinosaur speciation rates prior to the K-Pg boundary. The second, reviewed by Sarjeant & Currie [11], Barrett *et al.* [12] and Archibald [13], suggests a decline on the timescale of hundreds of thousands or several millions of years related to non-bolide factors such as Deccan flood volcanism [10,14,15]. Finally, and most recently, Sakamoto *et al.* [16] proposed that dinosaurs were in global decline with falling speciation rates up to 35 Myr before the end-Cretaceous extinction and that non-avian dinosaurs were on a long-term trajectory toward extinction regardless of other external environmental factors. Nevertheless, proponents of all three hypotheses found evidence for continued high speciation rates among ceratopsid and hadrosaurid ornithischians in the latest Cretaceous compared to those in other dinosaur groups [3,16].

Prior to the bolide impact, the Late Cretaceous was a turbulent time in Earth history, featuring vast effusive volcanic eruptions [17] and major changes in global temperature and sea level [18–22]. Although it is possible that these environmental changes would have impacted long-term diversity trends in dinosaurs—on both global and local scales—the Mesozoic dinosaur record is too patchily distributed in time and space to provide the resolution necessary to test many of these hypotheses [3]. Moreover, a recent niche-modelling approach has indicated that changing climates might have had far less influence on non-avian dinosaur extinction than previously thought [10].

Reconstruction of past diversity patterns is difficult, as the 'raw' patterns of taxonomic diversity gleaned from the fossil record are subject to many hierarchical biases, ranging from initial fossilization potential, through the availability of fossiliferous rock outcrop, to anthropic collection biases (e.g. [23–28], also see [29]). Different methods have been applied in attempts to correct for these factors, including subsampling techniques (rarefaction, shareholder quorum subsampling and TRiPs) and comparisons of diversity with collections-related proxies (e.g. amount of outcrop area, numbers of fossil-bearing localities), with substantial disagreement over the patterns recovered (e.g. [3,8,9,12,30–32]).

Sakamoto *et al.* [16] proposed a new approach for examining the diversity dynamics of Late Cretaceous dinosaurs, eschewing the traditional approaches based on taxon counting and focusing instead on speciation rates across dinosaur phylogeny. They extracted the node count, i.e. the number of nodes from the root to the tip, and time elapsed, i.e. the root to tip distance in Myr, for each taxon included within two published dinosaur supertrees [33,34]. The authors then investigated how node count changed through time using three models: (i) a null model, where node count was modelled as a linear function of the time elapsed from root to tip (figure 1*a*), indicating no slowdown in speciation rate; (ii) a slowdown to asymptote model, where node count was modelled as a function of the square root of time elapsed from root to tip (figure 1*b*), indicating an initial reduction in the speciation rate prior to levelling off to a stable level, and (iii) a downturn or speciation slowdown model, where node count was modelled as a function of time elapsed from root to tip and its quadratic term (figure 1*c*), indicating a continual downturn in speciation rates. Sakamoto *et al.* [16] found evidence for the downturn model (figure 1*c*) for dinosaurs as a whole and for the majority of dinosaur sub-clades,

with the exceptions of Hadrosauriformes and Ceratopsidae. They concluded that dinosaurs experienced a long-term decline in speciation rates throughout the Cretaceous prior to the extinction of all non-avian taxa at the K-Pg boundary, with the onset of this decline occurring approximately 100 Ma in the 'mid'-Cretaceous, up to 35 Myr prior to the bolide impact.

However, several lines of evidence undermine this conclusion. First, Hadrosauriformes and Ceratopsidae (representing approximately 14% of all genera included in their study) showed rapid species proliferation throughout the Late Cretaceous, suggesting that not all dinosaurs were declining prior to the K-Pg boundary [16] (see also [3]). Second, the level of support for the downturn model varied depending on which supertree was used and how the supertrees were dated, with the Lloyd et al. [34] supertree best fitted by a slowdown to asymptote model in most analyses rather than a downturn [16]. Third, the two dinosaur supertrees used in the analyses were based on phylogenetic hypotheses constructed prior to 2013 [33,34]. Since then there has been extensive phylogenetic work on various dinosaur sub-clades that were undersampled in these supertrees (e.g. [35–37]), and it is possible that the inclusion of these taxa may affect the results of these analyses. Fourth, some of the methodological choices made in fitting and selecting the models may have been intrinsically biased towards selecting the downturn model (see Materials and methods). Finally, ecological niche modelling suggests that the amount of habitable area available to Late Cretaceous dinosaurs did not decline during this interval (and may have increased), raising the possibility that these communities are taxonomically undersampled and were probably richer than currently recognized [10].

Here, we test whether non-avian dinosaurs were in long-term decline prior to the K-Pg boundary by using recently published dinosaur phylogenies and Bayesian phylogenetic generalized linear mixed models (GLMMs) to assess their fit to the three speciation models discussed above (figure 1). We do not find strong support for the downturn model in our analyses, which suggests that dinosaurs were not declining prior to the K-Pg boundary or, if they were, we cannot confidently detect this signal with phylogenetic data alone.

# 2. Materials and methods

## 2.1. Phylogenetic trees

Sakamoto et al. [16] used the dinosaur supertrees of Benson et al. [33] (number of taxa [n] = 614) and Lloyd et al. [34] (n = 420), including two subtly different versions of the Benson et al. [33] supertree, which reflect differences in hypotheses of sauropod inter-relationships. Each of these trees was dated using either the midpoint date, first occurrence date (FAD) or last occurrence date (LAD) for each taxon taken from Benson et al. [33] and Lloyd et al. [34], resulting in nine dated trees. We used all of these trees to enable comparisons between our results and those presented by Sakamoto et al. [16].

We expanded this comparison by using an additional set of nine recently published non-avian dinosaur phylogenies that include representation of all the major clades present during the late Mesozoic, including thyreophorans [37–39] (n = 23, 50 and 57, respectively), ceratopsians [40] (n = 27 and 30, respectively), hadrosauriforms [41] (n = 62), sauropods [36,42] (n = 87 and 76, respectively) and theropods [35] (n = 141). We used TNT v. 1.5 [43] with the search settings of the original analysis to produce consensus trees from each (see electronic supplementary material, Methods for more details). We used the same methods used in these papers to infer the trees, as our aim was to recreate these published trees, not to provide new phylogenetic inferences. These trees were then time-scaled using the R package paleotree v. 3.1.0 [44], incorporating uncertainty by sampling taxon ages randomly from a uniform distribution between their maximum and minimum possible ages. We chose this protocol for consistency with the dated trees in Sakamoto et al. [16]. Maximum and minimum possible ages came from the Paleobiology Database (data downloaded 2 April 2019 using the group names 'Ornithischia', 'Sauropoda' and 'Theropoda', and the following parameters: Taxonomic rank = genera and below, Taxonomic status = accepted names and junior synonyms, Additional output blocks = age range overall). Zero-length branches were lengthened by imposing a minimum branch duration of 1 Myr; [45] (see electronic supplementary material for analyses ensuring that this procedure did not bias the root ages of the trees towards being too old). Because this procedure randomly samples taxon ages, we exported 100 dated trees for each tree and ran our models (see below) on each. In total, we used 900 trees (nine trees each dated 100 times) in our analyses, plus the nine trees used by Sakamoto et al. [16].

## 2.2. Node counts and time elapsed

We extracted node counts for each taxon in each of our 909 trees using the 'nodepath' function within the 'ape' R package [46]. Node count is the number of nodes from the root to the tip for each taxon. Time elapsed (Myr) was extracted using the 'picante' R package [47]. Time elapsed is the root to tip distance for each taxon.

## 2.3. Model fitting

We used the same three models described in Sakamoto *et al.* [16] and discussed in the Introduction. These were (i) the null model, with node count modelled as a linear function of time elapsed from root to tip (figure 1*a*; nodecount = f(time elapsed)), (ii) the slowdown to asymptote model, with node count modelled as a function of the square root of time elapsed from root to tip (figure 1*b*; nodecount = f($\sqrt{}$time elapsed)), and (iii) the downturn model, with node count modelled as a function of time elapsed from root to tip and its quadratic term (figure 1*c*; nodecount = f(time elapsed + time elapsed$^2$)). Defining the theoretical value for the node count when no time has elapsed is not straightforward. The trees have no tips at $t_0$, meaning that node count will be zero; however, because the tree has a root node, there is technically a node count of 1. We therefore fitted models where the intercept is estimated (following Sakamoto *et al.* [16]), with the intercept set to zero, and with the intercept set to 1.0, to determine how this influenced our results.

We then fitted the models as Bayesian phylogenetic GLMMs in the R package MCMCglmm [48]. The response variable, node count, is a count so models were fitted with Poisson errors. As closely related species must have more similar node count values than distantly related species, we included the phylogeny (as the inverse of the phylogenetic variance-covariance matrix) as a random effect to account for this phylogenetic autocorrelation. We ran each MCMCglmm model for $5 \times 10^5$ iterations sampling at every 1000 iterations and discarding the first $5 \times 10^4$ iterations as burn-in. We used the default priors for MCMCglmm ($\mu = 0$ and $V = I \times 10^{10}$ for fixed effects and parameter expanded priors, and $V = 1$, $v = 1$, $\alpha\mu = 0$ and $\alpha V = 25^2$ for the phylogenetic random effects). All models had a mean effective sample size (ESS; estimated using the R package coda; [49]) of greater than 200.

We fitted each of the three models to all of our 909 trees (figure 1), with intercepts set either to zero (see above) or with estimated intercepts (following Sakamoto *et al.* [16]), or with intercepts set to 1.0 (see above). For each model, we extracted the deviance information criterion (DIC) and used this to identify the 'best' model for each tree, defined as the model with the smallest DIC value, and a difference in DIC of 4 units. Models with less than 4 units difference were not considered to be different. Note that this differs from the procedure of Sakamoto *et al.* [16], which used the significance of the model parameters (time, time elapsed$^2$ or $\sqrt{}$time elapsed) to choose between models where the difference in DIC was less than 4 units. We feel that our procedure is a potentially fairer test between models, as quadratic terms are almost always significant and would thus lead to the downturn model (figure 1*c*) being preferentially selected over either the null (figure 1*a*) or slowdown to asymptote models (figure 1*b*) even in cases where model fit was similar. For models with estimated intercepts, we also extracted the posterior means of the intercepts to examine how these varied.

All analyses used R v. 3.6 [50] and reproducible scripts are available on GitHub (https://github.com/nhcooper123/dino-trees/ [51]). The data required to rerun our analyses can be found in the NHM Data Portal (https://dx.doi.org/10.5519/0034257) [52].

# 3. Results

The best model varied between trees, between differently dated versions of the same tree, and on the basis of whether intercepts were estimated, set to zero, or set to 1.0 (figures 2 and 3; electronic supplementary material, tables S1–S3 and figures S1–S4). In models where intercepts were estimated (figures 2 and 3; electronic supplementary material table S1), 100 (11%) of our 900 new trees unambiguously favoured the downturn model, none unambiguously favoured the slowdown to asymptote model and 24 (3%) favoured either the downturn or the slowdown to asymptote model. No trees favoured the null model and 776 (86%) did not favour any model at all (figure 3). The Lloyd *et al.* [34] supertree and both versions of the Benson *et al.* [33] supertree favoured either the downturn or the slowdown to asymptote model (figure 3; electronic supplementary material, table S1). The intercepts of these models varied between zero and 2.5 (electronic supplementary material figure S5).

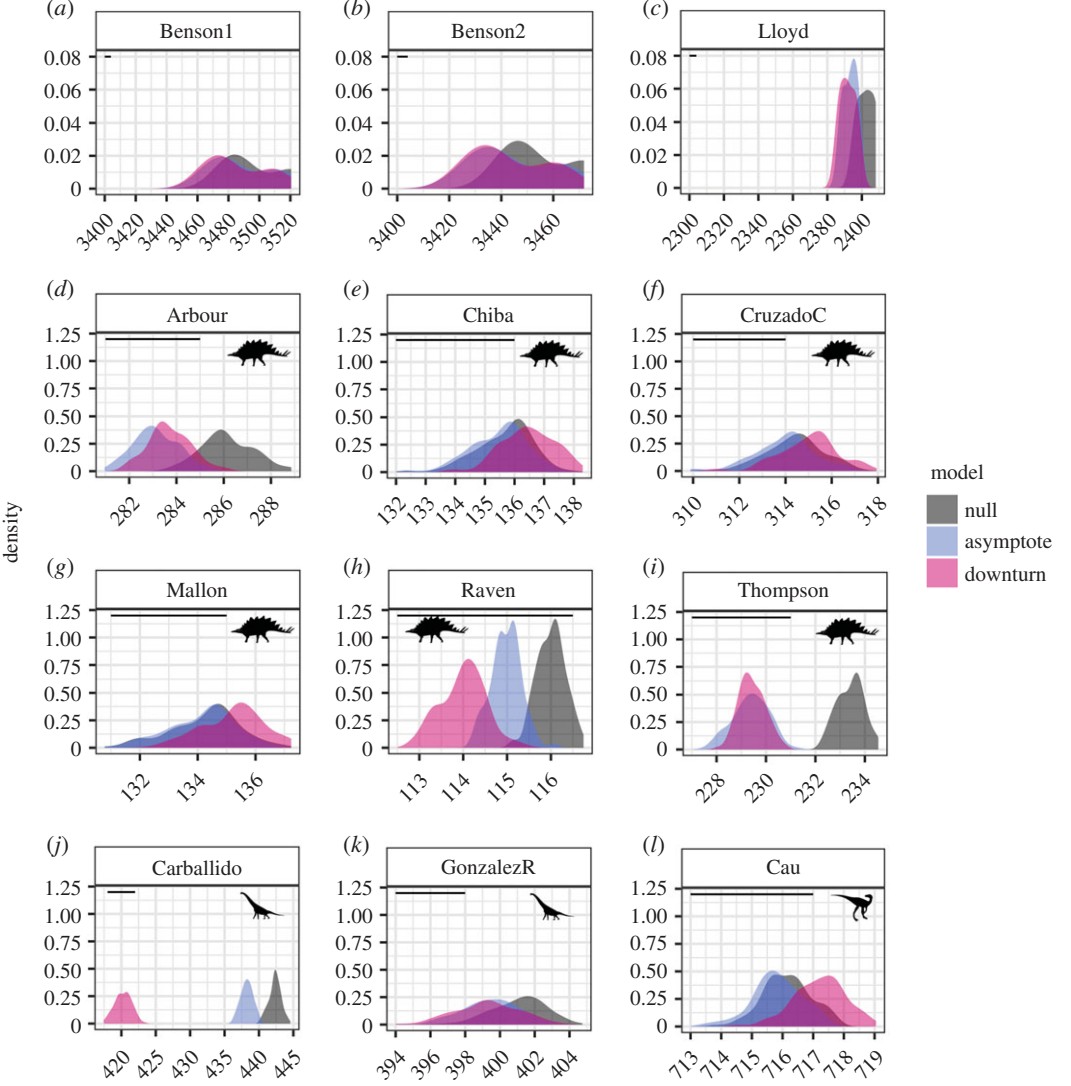

**Figure 2.** (*a–l*) DIC for the three models (figure 1) for each of 900 trees in this study, plus nine trees from Sakamoto *et al.* [16] with intercepts estimated. Horizontal lines show the length of 4 DIC units (note that the *x*-axis differs on each plot), the difference required for one model to be preferred over another. The *y*-axis is smaller in the first three panels as these show results from models fitted to three trees only whereas the later panels used 100 trees (see text). Panels are ordered based on the dinosaur group in each tree as follows Dinosauria: [33] (two trees) and [34] Ornithischia: [37–41,53], Sauropodomorpha: [36,42], Theropoda: [35]. Silhouettes are from PhyloPic.org: Ornithischia by Andrew A. Farke; Sauropodomorpha by Scott Hartman; Theropoda by Marmelad.

Our analyses where intercepts were estimated produced qualitatively identical best model results for the Benson *et al.* [33] and Lloyd *et al.* [34] trees (see electronic supplementary material, table S4 and S01 in Sakamoto *et al.* [16]), indicating that the differences between our results and theirs were not due to errors in interpreting their methods.

For models where intercepts were set to zero (electronic supplementary material, figures S1 and S2; table S2), 207 (23%) of our 900 new trees unambiguously favoured the downturn model, 186 (21%) unambiguously favoured the slowdown to asymptote model and 472 (52%) favoured either the downturn or the slowdown to asymptote model. No trees favoured the null model and 35 (4%) favoured no model at all (electronic supplementary material figure S1; and table S2). All three Lloyd *et al.* [34] supertrees unambiguously favoured the slowdown to asymptote model, as did three of the Benson *et al.* [33] supertrees. One Benson *et al.* [33] supertree unambiguously favoured the downturn model and the remaining two Benson *et al.* [33] supertrees favoured either the downturn or the slowdown to asymptote model (electronic supplementary material figure S2; and table S2).

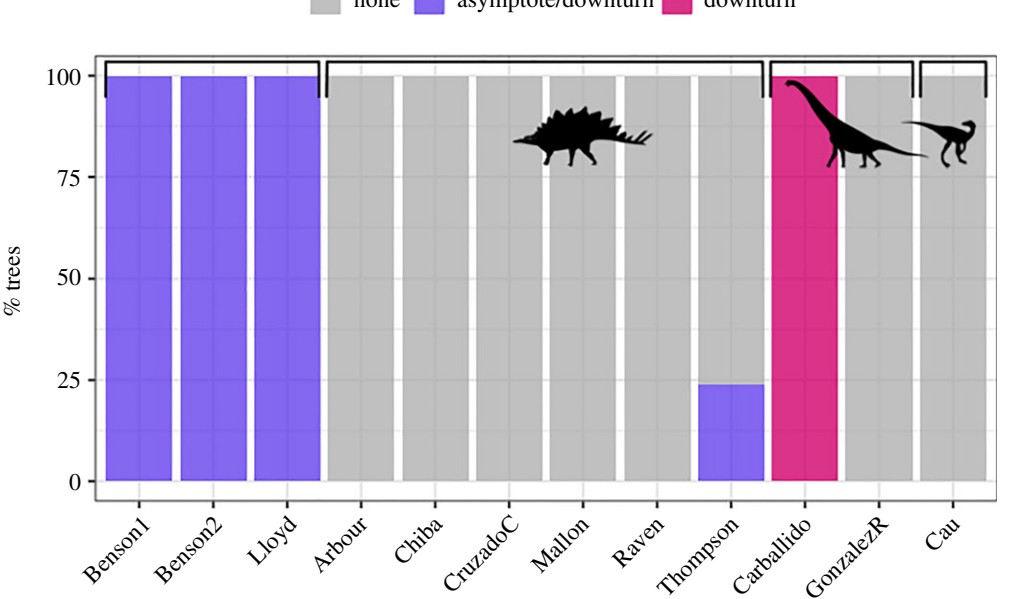

**Figure 3.** The 'best' model (figure 1) based on ΔDIC greater than 4 units for each of 900 trees in this study, plus nine trees from Sakamoto *et al.* [16], with intercepts estimated. Bars are grouped based on the dinosaur group in each tree as follows. Dinosauria: [33] (two trees) and [34]. Ornithischia: [37–41,53], Sauropodomorpha: [36,42], Theropoda: [35]. Silhouettes are from PhyloPic.org: Ornithischia by Andrew A. Farke; Sauropodomorpha by Scott Hartman; Theropoda by Marmelad.

For models where intercepts were set to 1.0 (electronic supplementary material, figures S3 and S4 and table S3), 183 (20.3%) of our 900 new trees unambiguously favoured the downturn model, 209 (23.2%) unambiguously favoured the slowdown to asymptote model and 469 (52.1%) favoured either the downturn or the slowdown to asymptote model. No trees favoured the null model and 39 (4.33%) favoured no model at all (electronic supplementary material figure S3; and table S3). All three Lloyd *et al.* [34] supertrees unambiguously favoured the slowdown to asymptote model, as did three of the Benson *et al.* [33] supertrees. One Benson *et al.* [33] supertree unambiguously favoured the downturn model and the remaining two Benson *et al.* [33] supertrees favoured either the downturn or the slowdown to asymptote model (electronic supplementary material figure S4; and table S3).

## 4. Discussion

In general, our results agree with those of Sakamoto *et al.* [16] but we disagree in our interpretation of those results. As described above, Sakamoto *et al.* [16] selected their best models based on a DIC number of at least 4 units difference. If, however, two models had similar DIC values they selected the best of these using the significance of the model parameters (time, time elapsed$^2$ or $\sqrt{\text{time}}$ elapsed). Because these quadratic terms were generally significant this led to their favouring the downturn model over the other two models in these ambiguous cases, potentially leading them to overstate the success of the downturn model. While this is a valid methodological choice, and differences in opinion about model selection procedures are common, we argue that as their selection of the downturn model as the 'best' model was methodologically equivocal it is unfair to say there is 'overwhelming support' for a downturn in dinosaur speciation rates prior to the Late Cretaceous [16, p. 5036]. Our conclusions also appear to be sensitive to the choice, or estimation, of intercepts in the models. Here, we advocate caution in interpreting the results of such models, as they may not accurately reflect the complexities of the underlying data.

Using the various combinations of trees, dating protocols and intercept assumptions listed above, we fitted a total of 2727 speciation models. Of these, only 518 (approx. 19%) unambiguously favoured the downturn model (figure 1*c*) that would be indicative of a continual downturn in dinosaur speciation rates before the K-Pg boundary. In particular, the sauropod tree of Carballido *et al.* [36] consistently favoured the downturn model, perhaps due to the choice and chronostratigraphic range of the taxa included in the study. However, even if we consider the 978 (approx. 36%) results where the

downturn and the slowdown to asymptote model were equally well supported as indicative of the downturn model alone, just over half of our models (approx. 55%) support the hypothesis that dinosaurs were in terminal decline during the latest Cretaceous or the rest of the Mesozoic Era. Instead, our results suggest that although some groups of dinosaurs may have experienced declining speciation rates before the K-Pg boundary, others did not, and it was not the prevalent pattern within the clade. Hence, we are sceptical that the roots of non-avian dinosaur extinction were based in the mid-Cretaceous (*contra* [16]), but suggest instead that dinosaur diversity would have remained high through the Late Cretaceous although taxon richness would have varied on a clade-by-clade basis.

A recent study by Chiarenza *et al.* [10] provides further support to our conclusions on the basis of climatic and ecological niche modelling. These authors found that the amount of habitable space for North American dinosaurs did not decline during the Late Cretaceous (and may, in fact, have increased slightly) but that the amount of fossiliferous rock outcrop available for this extended region is relatively small. Hence, they concluded that these communities are probably undersampled and were probably much richer than currently thought [10]. Their conclusion is also supported by other analyses that suggest small-bodied dinosaurs in particular are poorly sampled, even in those Late Cretaceous faunas that are thought to be the best-known [54]. In addition, other work on dinosaur morphological disparity and taxon richness finds that although some clades decreased in diversity at this time others maintained or increased diversity [3,55]. These palaeoecological, taphonomic and diversity-based approaches complement our phylogenetically based results and suggest an emerging consensus that supports the long-term maintenance of Late Cretaceous dinosaur richness, rather than models of extended decline. Our results emphasize the fact that the dinosaur fossil record suffers from numerous gaps and biases, and that any apparent decline in diversity could be due to systemic sampling errors. Nevertheless, even among these broadly consilient results, differences in reported pattern still require investigation. For example, Brusatte *et al.* [3] and Dean *et al.* [28] rejected a global long-term decline prior to the K-Pg boundary but found evidence for declines in species-richness and disparity at the end of the Cretaceous, Lloyd [30] found a long-term decline in sauropod and ornithischian diversity through the Cretaceous, and Barrett *et al.* [12] recovered evidence for negative trends in theropod and ornithischian diversity during the last two stages of the Cretaceous but identified a radiation of Late Cretaceous sauropodomorphs. These differing results highlight the need to increase the density and spatio-temporal scope of taxon sampling for these clades and to combine detailed, stratigraphically controlled, regional-level analyses to provide a truly holistic picture of the biotic changes that occurred during this critical interval.

Finally, we posit that although phylogenies can be very useful in resolving long-running evolutionary debates [56,57], they may not always solve all of the problems that remain. For example, a recent study found no correlation between rates of dinosaur morphological evolution and extinction, which suggests that analyses based on existing phylogenetic datasets might not be useful for addressing the question of non-avian dinosaur extinction [58]; however, this result may also indicate inaccuracies in the calculation of phenotypic diversification rates. In addition, the quality of the fossil record is generally poor for many groups, as it is spatially and temporally patchy, and gives a biased reflection of reality (e.g. [59,60]). For example, querying the Paleobiology Database (PBDB, accessed on 16 June 2020, using the group name 'Dinosauria' and time intervals = Cretaceous) shows a steep drop in sampled dinosaur genera from the Cenomanian (approx. 100 Ma) through to the end of the Santonian (approx. 83.6 Ma). This trend is relatively unstudied and represents a global lack of terrestrial fossil-bearing localities from this time that is due to high sea-level stands (see also [9,12]); such gaps might be difficult to fill due to genuinely low opportunities for fossil recovery. Conversely, the past three decades have witnessed a dramatic increase in the rates of new dinosaur species being discovered annually (with more than 50 per year since the 2000s; [55]), improvements in the accuracy of stratigraphical dating techniques [61] and the application of new analytical techniques that have enabled modern researchers to refine their taxonomic methods. We may never know the true levels of speciation and extinction of Mesozoic dinosaurs, but an increased focus on filling gaps in the fossil record will be the primary way in which palaeontologists will continue to build a more accurate picture of past dinosaur diversity.

Data accessibility. All data required to rerun our analyses can be found in the NHM Data Portal [52] (https://doi.org/10.5519/0034257), and reproducible scripts are available on GitHub (https://github.com/nhcooper123/dino-trees/ [51]).
Authors' contributions. J.A.B., P.M.B. and N.C. designed the study and wrote the manuscript; J.A.B. and T.J.R. collated the data; J.A.B. and N.C. ran the analyses, and all authors critically revised and approved the final manuscript.
Competing interests. We declare we have no competing interests.

Funding. J.A.B. is funded by a Leverhulme Trust PhD Research Grant (RL-2016-036) and Departmental Investment Funds (Earth & Life Sciences), Natural History Museum, London. T.J.R. is funded by a University of Brighton Science Scholarship.

Acknowledgements. Thanks to Manabu Sakamoto for sharing his R code, and two anonymous reviewers for their constructive comments. We thank the Willi Hennig society for enabling the free use of TNT.

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
