## [Reviewer comments · Royal Society Open Science]

Review History

RSOS-201195.R0 (Original submission)

Review form: Reviewer 1

Is the manuscript scientifically sound in its present form?

Yes

Are the interpretations and conclusions justified by the results?

Yes

Is the language acceptable?

Yes

Do you have any ethical concerns with this paper?

No

Have you any concerns about statistical analyses in this paper?

No

Recommendation?

Accept with minor revision (please list in comments)

Comments to the Author(s)

I support the publication of this work with minor revision. It presents quite an important set of analyses that probe the findings of Sakamoto et al (2016, PNAS) and add considerably to our ability to interpret those results. This is important because it relates to a long-standing debate about dinosaur decline in advance of the K-Pg extinction event. It also has importance in the development of phylogeny-based approaches to the study of diversification in the fossil record — this area has longer-term promise to unravel some key questions about macroevolutionary diversification in general. The paper has potential to be a classic in this area as it provides one of the most critical and reflective applications of these methods so far.

I wholeheartedly endorse publication in *Biology Letters*. It is certainly of appropriate significance and I would be disappointed if it was not published.

My key suggestions are as follows:

(1) The intercept of the models could be constrained to fall between 0 and 1.0, or set to 1.0 instead of 0. I believe that 1.0 nodes (i.e. the root node) should be present at even infinitesimally small values of time. Also, DIC could be used to evaluate models with constrained intercepts compared to those with estimated intercepts.

(2) Various minor suggestions about wording. Plus mentioning issues of sampling rate variation and potential solutions (even if not implemented here) in the introduction. They form an important theme of the discussion after all.

(3) Some relevant literature is not cited - you know, it's literature from my group so you can do eye-rolling if needed. I tried to restrict the amount that I suggested this however.

SPECIFIC COMMENTS

Title: "Evolutionary rates" can mean several different things. It may be more informative instead to write 'diversification rates'.

Lines 53-56: "Determining trends in Mesozoic dinosaur species-richness is a contentious issue and has been the focus of numerous previous studies (e.g. [3,8-10]). Many different hypotheses have been proposed, but there are currently two major schools of thought".

>Change "Determining trends in Mesozoic dinosaur species-richness" to "Determining trends in Mesozoic dinosaur species-richness *leading up to the K-Pg*". That would reflect the focus of this paragraph more accurately and avoid the misleading implication that studies of dinosaur species richness are restricted to opinions about the K-Pg.

>Also, I think it would be more instructive to suggest three different viewpoints here, because Sakamoto's study on its own does not reflect a 'major school of thought', whereas an actual 'major school of thought' is missing here. My view of the literature is that two things have been widely suggested: (1) No decline in advance of the boundary. (2) A decline on the timescale of 100,000s or millions of years related to non-bolide factors such as Deccan flood volcanism (older literature reviewed by Alvarez 1983 PNAS, and also see Dean et al 2020, cited in text, for more recent literature). (3) Sakamoto presented a third view that there was a decline on the timescale of 10s of millions of years without noting that no-one really had suggested this previously.

Lines 75-78: "Reconstruction of past diversity patterns is difficult, as the 'raw' patterns of taxonomic diversity gleaned from the fossil record are subject to many hierarchical biases, ranging from initial fossilization potential, through the availability of fossiliferous rock outcrop, to anthropic collection biases (e.g. [18-23])"

>Refs 18-23 include various potentially-appropriate citations about fossil record sampling heterogeneity. Some of them are classics (Signor & Lipps 1982 on edge effects). Some of them are relatively unfocussed studies that provide laundry lists of untested qualitative points. Vilhena and Smith (2013) is a study of spatial bias in the marine fossil record that doesn't really propose a solution to the problem.

>Work presenting new methods to address spatial-bias and applying that to the dinosaur fossil record is not cited.

Close RA, Benson RBJ, Alroy J, Carrano MT, Clearly T, Dunne E, Mannion PD, Uhen M, Butler RJ. 2020. The apparent exponential radiation of Phanerozoic land vertebrates reflects spatial sampling biases. *Proceedings of the Royal Society B* 287, 20200372.

>I also think it might be appropriate to cite Benson (2018, Dinosaur macroevolution and macroecology) which contains discussions and plots related to dinosaur diversity dynamics, spatial sampling variation, regional heterogeneity, Sakamoto et al and other topics that are discussed in the current paper. Some of the statements in Section 5 of Benson (2018), on several different topics, are quite similar to those of the current paper.

Lines 78-81: "Different methods have been applied in attempts to correct for these factors, including various subsampling techniques (rarefaction, Shareholder Quorum Subsampling and TRiPs) and comparisons of diversity with collections-related proxies"

>Spatial bias is emphasised in earlier text, but none of the methods written into the text here can correct for spatial bias. New approaches to address spatial bias were presented by Close et al (2020), which also report patterns of dinosaur diversity.

Lines 82-83: "...with substantial disagreement over the patterns recovered (e.g. [3,8,9,24-29])".

>Refs 27-29 are papers evaluating method performance rather than papers about patterns of dinosaur diversity. So they seem to be cited in the wrong place. Arguably, they don't need to be cited at all unless the text includes statements about method performance.

Lines 85-137: These paragraphs list potential problems with the analyses/interpretation of Sakamoto et al. There is another problem that is not mentioned in the text but is potentially very important. Fossil phylogenies are different to phylogenies of living species because times represent sampling events. Sampling events also influence the node counts, because each sampling event adds one tip during the sampled interval and one node during some older interval. Sakamoto et al didn't really do anything that would be appropriate to address this but a lot of progress was made on appropriate models by other groups – primarily Tanja Stadler's group and the fossilise-birth-death (FBD) model. Although it might not be tractable to apply that to this problem right now, I think it would be right to mention this shortcoming and that solutions might be possible in future using models such as the FBD model, even though that is not applied here.

Lines 143-145: "two subtly different versions of the Benson et al. [30] supertree, which reflect plausible differences in the relationships of several taxa"

>The trees reflect difference in hypotheses of sauropod inter-relationships. This could be stated instead of "relationships of several taxa".

Line 146: "first occurrence date (FAD) or last occurrence date (LAD)"

>These mostly aren't FADs and LADs. Most are the oldest-age and youngest-age bounds on uncertainty associated with taxon ages. So, the same as "Maximum and minimum possible ages" (lines 160-161).

Lines 190-191: "As closely related species are likely to have more similar node count values"

>By definition they -must- have similar node counts. If you agree, consider rephrasing 'are more likely to have'.

Line 228: "776 (86%) did not favour any model at all"

>I understand entirely why this is written. However, the statement doesn't contain much information. I'd really like to see the densities for the Lloyd et al and Benson et al. results but they aren't shown in Figs 2 or 4. There shouldn't be an issue with space-availability to include these because the panels aren't complex and their sizes could be reduced.

Figures 2 and 4.

>It would be useful to write out the name of the dinosaur subgroup on each panel. It's a bit tricky to figure it out as currently-presented. You can make it very easy for readers to see at-a-glance what groups are analysed.

Lines 296-298: "Their conclusions might also have been skewed by reliance on estimated intercepts [11] rather than zero-intercepts, as the latter would, perhaps, be more biologically plausible (as there should be no new nodes at t0)"

>I agree that setting the intercept to zero has a big effect on the result - that's clear from the analyses. There are a few things to say on this. Firstly: I'm not sure how many nodes there should be at $t = 0$, because the count of nodes at $t = 0$ does not have a straightforward interpretation. In reality, at $t > 0$, even at infinitesimally small values, at least 1.0 nodes must have occurred because $t = 0$ is the time of the most basal node in the tree. The fact that infinitesimally small values are essentially equal to zero is important and suggest that when time = 0, the node count should be 1.0. However, zero tips have occurred at time = 0, so in reality we haven't had any data-items that would record time = 0 and nodes = 1. By definition, zero tips occur until time = 1 (because of the algorithm use to scale the trees to time). By time = 1 all we know is that node counts are greater than or equal to 1.0.

>These are my recommendations:

>(1) Set the intercept to 1.0 instead of or as well as 0. Or constrain the value of the intercept to be between 0 and 1.0 but don't specify it precisely.

>(2) Use DIC to evaluate which model is best (i) a model in which the intercept is entirely unconstrained, (ii) a model in which the intercept is between 0 and 1.0, (iii) any other intercept setting that you like.

Lines 297-298: "would, perhaps, be more biologically plausible"

>"perhaps" — it's very British to write this way. But it's also quite vague. This isn't a matter of 'biological plausibility'. In fact, it is axiomatic in the analysis. So I think you should include much stronger statements about this.

>The methodological decisions would also be clearer to me if the estimated intercept values were reported in a graph or similar. If the argument is about 'biological plausibility', so this is quite important. Some values of the intercept are presumably more plausible than others (e.g. a value of 1.0 is quite plausible, see above).

Lines 311: "support the idea that dinosaurs were in terminal decline:

>Change to "support the hypothesis that dinosaurs were in terminal decline"

Lines 334-337: "Our results, and those of many others (e.g. [3,9,10,23,50]), emphasize the fact that the dinosaur fossil record suffers from numerous gaps and biases, and that any apparent decline in diversity could be due to systemic sampling errors"

>This may be the view of the authors (and is also my view - I agree with you). However, it is not supported by any of the analyses presented here, which don't contain any test of the effect of sampling rate variation. See also my note on the Introduction (above) - lines 85-137 do not cite sampling rate variation as a potential issue with this type of analysis. However, that does seem to be the view of the current authors.

Lines 351-356: "For example, a recent study found no correlation between rates of dinosaur morphological evolution and extinction, which suggests that analyses based on existing phylogenetic datasets might not be useful for addressing the question of non-avian dinosaur extinction [54]"

>It isn't clear why that result would lead to the conclusion stated here unless we assumed a priori that diversification rates must be correlated with phenotypic rates. Even if we did assume that, then the result might imply that the phenotypic rates are inaccurate (likely, given what they are based on).

Review form: Reviewer 2 (Martín Ezcurra)

Is the manuscript scientifically sound in its present form?

Yes

Are the interpretations and conclusions justified by the results?

Yes

Is the language acceptable?

Yes

Do you have any ethical concerns with this paper?

No

Have you any concerns about statistical analyses in this paper?

Yes

Recommendation?

Accept with minor revision (please list in comments)

Comments to the Author(s)

The manuscript of Bensor and colleagues seek to address on the main questions regarding the extinction of non-avian dinosaurs, were evolutionary rates decelerating or very low before their extinction? Their main goal is to test the results of a recently published paper that found a downturn of evolutionary rates for most dinosaur clades close to the end of the Cretaceous Period (Sakamoto et al., 2016). I consider that the manuscript is very well written, the figures are very informative, and the conclusions built on well supported results. I haven't found major issues in the manuscript and I have only some moderate comments, but they may imply further analyses or to re-run some analyses (see below). I also made some minor comments and suggestions in an edited version of the manuscript (Appendix A). As a result, I think that the manuscript requires a minor to moderate revision before acceptance.

Comments:

- I understand that you haven't used the stochastic dating method 'cal3'. Can you briefly explain why you haven't used it and you prefer a partially stochastic dating but setting a fixed minimum branch length?
- You have used a minimum branch length of 1 million years, which it is OK for most datasets. However, if you have large trees, it may result in artificially very old root ages. Can you report the mean and standard deviation of the root ages of each of the eighteen trees (9 new trees and 9 trees from Sakamoto et al. 2016, isn't?). After doing that, please, evaluate if the root ages are compatible with the inferred origin of the clades (e.g. that the origin of Theropoda is not occurring in the Permian). If the latter is the case for any of the trees/clades, I suggest using a shorter minimum branch length or fixing the root of the tree using the inferred minimum divergence time between clades.
- Different minimum branch lengths may have strong impacts in the results. Thus, I would like to see how sensitive are the results to calibrations using at least two other minimum branch lengths, e.g. 0.1 myr and 5 myr.
- In the supplementary information you state that you have used a 50% majority rule consensus tree for some analyses (e.g. Arbour et al. 2016). There is no reason to exclude some topologies only because they occur in less than 50% of the most parsimonious trees if you are running a cladistic analysis. So, use a strict consensus tree or a strict reduced consensus tree (with wildcard taxa pruned a posteriori) instead.
- Some of these taxon-character matrices are relatively large (>70 taxa). In some of them you have used the New Technology algorithms of TNT during the tree searches. However, 100 replicates of these algorithms may not generate a good sample of optimal results (e.g. Cau analysis). A more appropriate protocol is to use this combination of algorithms but until an a priori determined number of optimal results is reached. You can do this with the TNT command 'xmult = hits 100;'. I understand that you have followed the search protocols of the original papers, but you should be sure that the tree searches are properly done. It is the same for the consensus trees (see above). I suggest doing searches using the New Technology algorithms if you have more than 70 taxa.
- I can't see a good reason to exclude taxa before the time calibrations. Can you explain why you are doing that in a few analyses?

Decision letter (RSOS-201195.R0)

Dear Mr Bonsor

On behalf of the Editors, we are pleased to inform you that your Manuscript RSOS-201195 "Dinosaur evolutionary rates were not in decline prior to the K-Pg boundary" has been accepted for publication in Royal Society Open Science subject to minor revision in accordance with the referees' reports. Please find the referees' comments along with any feedback from the Editors below my signature.

Note that Reviewer 1 makes a slightly confusing reference to Biology Letters -- please assume that the Reviewer in fact means RSOS.

Please submit your revised manuscript and required files (see below) no later than 7 days from today's (ie 21-Sep-2020) date. Note: the ScholarOne system will 'lock' if submission of the revision is attempted 7 or more days after the deadline. If you do not think you will be able to meet this deadline please contact the editorial office immediately.

on behalf of Dr Julia Brenda Desojo (Associate Editor) and Peter Haynes (Subject Editor)
openscience@royalsociety.org

Reviewer comments to Author:

Reviewer: 1
Comments to the Author(s)

I support the publication of this work with minor revision. It presents quite an important set of analyses that probe the findings of Sakamoto et al (2016, PNAS) and add considerably to our ability to interpret those results. This is important because it relates to a long-standing debate about dinosaur decline in advance of the K-Pg extinction event. It also has importance in the development of phylogeny-based approaches to the study of diversification in the fossil record — this area has longer-term promise to unravel some key questions about macroevolutionary diversification in general. The paper has potential to be a classic in this area as it provides one of the most critical and reflective applications of these methods so far.

I wholeheartedly endorse publication in Biology Letters. It is certainly of appropriate significance and I would be disappointed if it was not published.

My key suggestions are as follows:

(1) The intercept of the models could be constrained to fall between 0 and 1.0, or set to 1.0 instead of 0. I believe that 1.0 nodes (i.e. the root node) should be present at even infinitesimally small values of time. Also, DIC could be used to evaluate models with constrained intercepts compared to those with estimated intercepts.

(2) Various minor suggestions about wording. Plus mentioning issues of sampling rate variation and potential solutions (even if not implemented here) in the introduction. They form an important theme of the discussion after all.

(3) Some relevant literature is not cited - you know, it's literature from my group so you can do eye-rolling if needed. I tried to restrict the amount that I suggested this however.

SPECIFIC COMMENTS

Title: "Evolutionary rates" can mean several different things. It may be more informative instead to write 'diversification rates'.

Lines 53-56: "Determining trends in Mesozoic dinosaur species-richness is a contentious issue and has been the focus of numerous previous studies (e.g. [3,8-10]). Many different hypotheses have been proposed, but there are currently two major schools of thought".

>Change "Determining trends in Mesozoic dinosaur species-richness" to "Determining trends in Mesozoic dinosaur species-richness *leading up the the K-Pg*". That would reflect the focus of this paragraph more accurately and avoid the misleading implication that studies of dinosaur species richness are restricted to opinions about the K-Pg.

>Also, I think it would be more instructive to suggest three different viewpoints here, because Sakamoto's study on its own does not reflect a 'major school of thought', whereas an actual 'major school of thought' is missing here. My view of the literature is that two things have been widely suggested: (1) No decline in advance of the boundary. (2) A decline on the timescale of 100,000s or millions of years related to non-bolide factors such as Deccan flood volcanism (older literature reviewed by Alvarez 1983 PNAS, and also see Dean et al 2020, cited in text, for more recent literature). (3) Sakamoto presented a third view that there was a decline on the timescale of 10s of millions of years without noting that no-one really had suggested this previously.

Lines 75-78: "Reconstruction of past diversity patterns is difficult, as the 'raw' patterns of taxonomic diversity gleaned from the fossil record are subject to many hierarchical biases, ranging from initial fossilization potential, through the availability of fossiliferous rock outcrop, to anthropic collection biases (e.g. [18-23])"

>Refs 18-23 include various potentially-appropriate citations about fossil record sampling heterogeneity. Some of them are classics (Signor & Lipps 1982 on edge effects). Some of them are relatively unfocussed studies that provide laundry lists of untested qualitative points. Vilhena and Smith (2013) is a study of spatial bias in the marine fossil record that doesn't really propose a solution to the problem.

>Work presenting new methods to address spatial-bias and applying that to the dinosaur fossil record is not cited.

Close RA, Benson RBJ, Alroy J, Carrano MT, Clearly T, Dunne E, Mannion PD, Uhen M, Butler RJ. 2020. The apparent exponential radiation of Phanerozoic land vertebrates reflects spatial sampling biases. *Proceedings of the Royal Society B* 287, 20200372.

>I also think it might be appropriate to cite Benson (2018, Dinosaur macroevolution and macroecology) which contains discussions and plots related to dinosaur diversity dynamics, spatial sampling variation, regional heterogeneity, Sakamoto et al and other topics that are discussed in the current paper. Some of the statements in Section 5 of Benson (2018), on several different topics, are quite similar to those of the current paper.

Lines 78-81: "Different methods have been applied in attempts to correct for these factors, including various subsampling techniques (rarefaction, Shareholder Quorum Subsampling and TRiPs) and comparisons of diversity with collections-related proxies"

>Spatial bias is emphasised in earlier text, but none of the methods written into the text here can correct for spatial bias. New approaches to address spatial bias were presented by Close et al (2020), which also report patterns of dinosaur diversity.

Lines 82-83: "...with substantial disagreement over the patterns recovered (e.g. [3,8,9,24-29])".

>Refs 27-29 are papers evaluating method performance rather than papers about patterns of dinosaur diversity. So they seem to be cited in the wrong place. Arguably, they don't need to be cited at all unless the text includes statements about method performance.

Lines 85-137: These paragraphs list potential problems with the analyses/interpretation of Sakamoto et al. There is another problem that is not mentioned in the text but is potentially very important. Fossil phylogenies are different to phylogenies of living species because times represent sampling events. Sampling events also influence the node counts, because each sampling event adds one tip during the sampled interval and one node during some older interval. Sakamoto et al didn't really do anything that would be appropriate to address this but a lot of progress was made on appropriate models by other groups — primarily Tanja Stadler's group and the fossilise-birth-death (FBD) model. Although it might not be tractable to apply that to this problem right now, I think it would be right to mention this shortcoming and that solutions might be possible in future using models such as the FBD model, even though that is not applied here.

Lines 143-145: "two subtly different versions of the Benson et al. [30] supertree, which reflect plausible differences in the relationships of several taxa"

>The trees reflect difference in hypotheses of sauropod inter-relationships. This could be stated instead of "relationships of several taxa".

Line 146: "first occurrence date (FAD) or last occurrence date (LAD)"

>These mostly aren't FADs and LADs. Most are the oldest-age and youngest-age bounds on uncertainty associated with taxon ages. So, the same as "Maximum and minimum possible ages" (lines 160-161).

Lines 190-191: "As closely related species are likely to have more similar node count values"

>By definition they -must- have similar node counts. If you agree, consider rephrasing 'are more likely to have'.

Line 228: "776 (86%) did not favour any model at all"

>I understand entirely why this is written. However, the statement doesn't contain much information. I'd really like to see the densities for the Lloyd et al and Benson et al. results but they aren't shown in Figs 2 or 4. There shouldn't be an issue with space-availability to include these because the panels aren't complex and their sizes could be reduced.

Figures 2 and 4.

>It would be useful to write out the name of the dinosaur subgroup on each panel. It's a bit tricky to figure it out as currently-presented. You can make it very easy for readers to see at-a-glance what groups are analysed.

Lines 296-298: "Their conclusions might also have been skewed by reliance on estimated intercepts [11] rather than zero-intercepts, as the latter would, perhaps, be more biologically plausible (as there should be no new nodes at t0)"

>I agree that setting the intercept to zero has a big effect on the result - that's clear from the analyses. There are a few things to say on this. Firstly: I'm not sure how many nodes there should be at $t = 0$, because the count of nodes at $t = 0$ does not have a straightforward interpretation. In reality, at $t > 0$, even at infinitesimally small values, at least 1.0 nodes must have occurred because $t = 0$ is the time of the most basal node in the tree. The fact that infinitesimally small values are essentially equal to zero is important and suggest that when time = 0, the node count should be 1.0. However, zero tips have occurred at time = 0, so in reality we haven't had any data-items that would record time = 0 and nodes = 1. By definition, zero tips occur until time = 1 (because of the algorithm use to scale the trees to time). By time = 1 all we know is that node counts are greater than or equal to 1.0.

>These are my recommendations:

>(1) Set the intercept to 1.0 instead of or as well as 0. Or constrain the value of the intercept to be between 0 and 1.0 but don't specify it precisely.

>(2) Use DIC to evaluate which model is best (i) a model in which the intercept is entirely unconstrained, (ii) a model in which the intercept is between 0 and 1.0, (iii) any other intercept setting that you like.

Lines 297-298: "would, perhaps, be more biologically plausible"

>"perhaps" — it's very British to write this way. But it's also quite vague. This isn't a matter of 'biological plausibility'. In fact, it is axiomatic in the analysis. So I think you should include much stronger statements about this.

>The methodological decisions would also be clearer to me if the estimated intercept values were reported in a graph or similar. If the argument is about 'biological plausibility', so this is quite important. Some values of the intercept are presumably more plausible than others (e.g. a value of 1.0 is quite plausible, see above).

Lines 311: "support the idea that dinosaurs were in terminal decline:

>Change to "support the hypothesis that dinosaurs were in terminal decline"

Lines 334-337: "Our results, and those of many others (e.g. [3,9,10,23,50]), emphasize the fact that the dinosaur fossil record suffers from numerous gaps and biases, and that any apparent decline in diversity could be due to systemic sampling errors"

>This may be the view of the authors (and is also my view - I agree with you). However, it is not supported by any of the analyses presented here, which don't contain any test of the effect of sampling rate variation. See also my note on the Introduction (above) - lines 85-137 do not cite sampling rate variation as a potential issue with this type of analysis. However, that does seem to be the view of the current authors.

Lines 351-356: "For example, a recent study found no correlation between rates of dinosaur morphological evolution and extinction, which suggests that analyses based on existing phylogenetic datasets might not be useful for addressing the question of non-avian dinosaur extinction [54]"

>It isn't clear why that result would lead to the conclusion stated here unless we assumed a priori that diversification rates must be correlated with phenotypic rates. Even if we did assume that, then the result might imply that the phenotypic rates are inaccurate (likely, given what they are based on).

Reviewer: 2

Comments to the Author(s)

The manuscript of Bensor and colleagues seek to address on the main questions regarding the extinction of non-avian dinosaurs, were evolutionary rates decelerating or very low before their extinction? Their main goal is to test the results of a recently published paper that found a downturn of evolutionary rates for most dinosaur clades close to the end of the Cretaceous Period (Sakamoto et al., 2016). I consider that the manuscript is very well written, the figures are very informative, and the conclusions built on well supported results. I haven't found major issues in the manuscript and I have only some moderate comments, but they may imply further analyses or to re-run some analyses (see below). I also made some minor comments and suggestions in an edited version of the manuscript. As a result, I think that the manuscript requires a minor to moderate revision before acceptance.

Comments:

- I understand that you haven't used the stochastic dating method 'cal3'. Can you briefly explain why you haven't used it and you prefer a partially stochastic dating but setting a fixed minimum branch length?

- You have used a minimum branch length of 1 million years, which it is OK for most datasets. However, if you have large trees, it may result in artificially very old root ages. Can you report the mean and standard deviation of the root ages of each of the eighteen trees (9 new trees and 9 trees from Sakamoto et al. 2016, isn't?). After doing that, please, evaluate if the root ages are compatible with the inferred origin of the clades (e.g. that the origin of Theropoda is not occurring in the Permian). If the latter is the case for any of the trees/clades, I suggest using a shorter minimum branch length or fixing the root of the tree using the inferred minimum divergence time between clades.

- Different minimum branch lengths may have strong impacts in the results. Thus, I would like to see how sensitive are the results to calibrations using at least two other minimum branch lengths, e.g. 0.1 myr and 5 myr.

- In the supplementary information you state that you have used a 50% majority rule consensus tree for some analyses (e.g. Arbour et al. 2016). There is no reason to exclude some topologies only because they occur in less than 50% of the most parsimonious trees if you are running a cladistic analysis. So, use a strict consensus tree or a strict reduced consensus tree (with wildcard taxa pruned a posteriori) instead.

- Some of these taxon-character matrices are relatively large (>70 taxa). In some of them you have used the New Technology algorithms of TNT during the tree searches. However, 100 replicates of these algorithms may not generate a good sample of optimal results (e.g. Cau analysis). A more appropriate protocol is to use this combination of algorithms but until an a priori determined number of optimal results is reached. You can do this with the TNT command 'xmult = hits 100;'. I understand that you have followed the search protocols of the original papers, but you should be sure that the tree searches are properly done. It is the same for the consensus trees (see above). I suggest doing searches using the New Technology algorithms if you have more than 70 taxa.

- I can't see a good reason to exclude taxa before the time calibrations. Can you explain why you are doing that in a few analyses?

===PREPARING YOUR MANUSCRIPT===

- one version identifying all the changes that have been made (for instance, in coloured highlight, in bold text, or tracked changes);
- a 'clean' version of the new manuscript that incorporates the changes made, but does not highlight them. This version will be used for typesetting.

===PREPARING YOUR REVISION IN SCHOLARONE===

-- Ensure that your data access statement meets the requirements at <https://royalsociety.org/journals/authors/author-guidelines/#data>. You should ensure that you cite the dataset in your reference list. If you have deposited data etc in the Dryad repository, please only include the 'For publication' link at this stage. You should remove the 'For review' link.

-- If you have uploaded ESM files, please ensure you follow the guidance at <https://royalsociety.org/journals/authors/author-guidelines/#supplementary-material> to include a suitable title and informative caption. An example of appropriate titling and captioning may be found at https://figshare.com/articles/Table_S2_from_Is_there_a_trade-off_between_peak_performance_and_performance_breadth_across_temperatures_for_aerobic_sc_ope_in_teleost_fishes_/3843624.

Author's Response to Decision Letter for (RSOS-201195.R0)

See Appendix B.

RSOS-201195.R1 (Revision)

Review form: Reviewer 1

Is the manuscript scientifically sound in its present form?

Yes

Are the interpretations and conclusions justified by the results?

Yes

Is the language acceptable?

Yes

Do you have any ethical concerns with this paper?

No

Have you any concerns about statistical analyses in this paper?

No

Recommendation?

Accept as is

Comments to the Author(s)

The paper should be accepted for publication

Review form: Reviewer 2 (Martín Ezcurra)

Is the manuscript scientifically sound in its present form?

Yes

Are the interpretations and conclusions justified by the results?

Yes

Is the language acceptable?

Yes

Do you have any ethical concerns with this paper?

No

Have you any concerns about statistical analyses in this paper?

No

Recommendation?

Accept as is

Comments to the Author(s)

I thank the authors to address each of the comments that I raised in the first version of the manuscript. Although I still think that it is very important to run a proper search of optimal trees, I understand the goal of the authors to obtain the same topologies as the original authors of the phylogenetic analyses (a good search goes beyond the expertise of the person in a specific taxonomic group). It will be very time consuming to re-run analyses using different search or consensus settings, so I think that the response of the authors to this criticism is justified under current circumstances.

Regarding the original comment of line 193, I wanted to know if you used half (above or below the diagonal) of the phylogenetic variance-covariance matrix. I think that it is important to allow replicating the analyses in the future.

I congratulate the authors for their work and I think that the manuscript can be accepted in its current form.

Decision letter (RSOS-201195.R1)

Dear Mr Bonsor,

It is a pleasure to accept your manuscript entitled "Dinosaur diversification rates were not in decline prior to the K-Pg boundary" in its current form for publication in Royal Society Open Science. The comments of the reviewer(s) who reviewed your manuscript are included at the foot of this letter.

Please ensure that you send to the editorial office an editable version of your accepted manuscript, and individual files for each figure and table included in your manuscript. You can send these in a zip folder if more convenient. Failure to provide these files may delay the

processing of your proof. You may disregard this request if you have already provided these files to the editorial office.

on behalf of Dr Julia Brenda Desojo (Associate Editor) and Peter Haynes (Subject Editor)
openscience@royalsociety.org

Associate Editor Comments to Author (Dr Julia Brenda Desojo):
Comments to the Author:

Both reviewers agree to accept this new complete version of the manuscript. Only one comment to be attended by the authors from R2:
Regarding the original comment of line 193, I wanted to know if you used half (above or below the diagonal) of the phylogenetic variance-covariance matrix. I think that it is important to allow replicating the analyses in the future.

Reviewer comments to Author:
Reviewer: 2

Comments to the Author(s)

I thank the authors to address each of the comments that I raised in the first version of the manuscript. Although I still think that it is very important to run a proper search of optimal trees, I understand the goal of the authors to obtain the same topologies as the original authors of the phylogenetic analyses (a good search goes beyond the expertise of the person in a specific taxonomic group). It will be very time consuming to re-run analyses using different search or consensus settings, so I think that the response of the authors to this criticism is justified under current circumstances.

Regarding the original comment of line 193, I wanted to know if you used half (above or below the diagonal) of the phylogenetic variance-covariance matrix. I think that it is important to allow replicating the analyses in the future.

I congratulate the authors for their work and I think that the manuscript can be accepted in its current form.

Reviewer: 1

Comments to the Author(s)

The paper should be accepted for publication

Appendix A**ROYAL SOCIETY
OPEN SCIENCE****Dinosaur evolutionary rates were not in decline prior to the
K-Pg boundary**

Journal:	Royal Society Open Science
Manuscript ID	RSOS-201195
Article Type:	Research
Date Submitted by the Author:	08-Jul-2020
Complete List of Authors:	Bonsor, Joseph; University of Bath, Biology and Biochemistry; Natural History Museum, Earth Sciences Barrett, Paul; Natural History Museum, Earth Sciences Raven, Thomas; Natural History Museum Cooper, Natalie; Natural History Museum, Life Sciences
Subject:	palaeontology < BIOLOGY, evolution < BIOLOGY, Palaeontology < EARTH SCIENCES
Keywords:	Dinosauria, Phylogeny, Bayesian, Speciation rates, K-Pg Boundary, GLMMs
Subject Category:	Earth and Environmental Science

Author-supplied statements

Relevant information will appear here if provided.

Ethics

Does your article include research that required ethical approval or permits?:

This article does not present research with ethical considerations

Statement (if applicable):

CUST_IF_YES_ETHICS :No data available.

Data

It is a condition of publication that data, code and materials supporting your paper are made publicly available. Does your paper present new data?:

Yes

Statement (if applicable):

All data required to rerun our analyses can be found in the NHM Data Portal (<https://doi.org/10.5519/0034257>), and all reproducible scripts are available on GitHub (<https://github.com/nhcooper123/dino-trees/>; DOI: 10.5281/zenodo.3935932.)

Conflict of interest

I/We declare we have no competing interests

Statement (if applicable):

CUST_STATE_CONFLICT :No data available.

Authors' contributions

This paper has multiple authors and our individual contributions were as below

Statement (if applicable):

JAB, PMB and NC designed the study and wrote the manuscript, JAB and TJR collated the data, JAB and NC ran the analyses, and all authors critically revised and approved the final manuscript.

**Dinosaur evolutionary rates were not in decline prior to the K-Pg**
**boundary**

**Joseph A. Bonsor^{1,2*}, Paul M. Barrett¹, Thomas J. Raven^{1,3} and Natalie Cooper⁴**

¹Department of Earth Sciences, Natural History Museum, London, Cromwell Road, London, SW7
5BD, UK.

²Department of Biology and Biochemistry, University of Bath, Claverton Down, Bath, BA2 7AY, UK.

³School of Environment and Technology, University of Brighton, Lewes Road, Brighton, BN2 4GA,
UK

⁴Department of Life Sciences, Natural History Museum, London, Cromwell Road, London, SW7 5BD,
UK.

*corresponding author

Suggested RH — Dinosaur evolutionary rates

**Abstract**

[revised manuscript text omitted]

*Figure 4: DIC for the three models (see Figure 1) for each of 900 trees with*
 *intercepts set to zero. Horizontal lines show the length of 4 DIC units (note that the*
 *x-axis differs on each plot), the difference required for one model to be preferred*
 *over another. Panels are ordered based on the dinosaur group in each tree as*
 *follows. Ornithischia: [34–38,49], Sauropodomorpha: [33,39], Theropoda: [32]*

Figure 5: The “best” model (see Figure 1) based on $\Delta DIC > 4$ units for each of 900
 trees in this study, plus nine trees from Sakamoto *et al.* 2016, with intercepts set to
 zero. Bars are grouped based on the dinosaur group in each tree as follows.

[revised manuscript text omitted]

380

381

**Author contributions**

JAB participated in the design of the study and drafted the manuscript; PMB
conceived of the study, designed the study, coordinated the study and helped draft
and critically revise the manuscript; TJR collected data and critically revised the
manuscript; NC conceived of the study, designed the study, coordinated the study
and helped draft and critically revise the manuscript. All authors gave final approval
for publication and agree to be held accountable for the work performed therein.

**Acknowledgements**

Thanks to Manabu Sakamoto for sharing his R code. JAB is funded by a Leverhulme
Trust PhD Research Grant (RL-2016-036) and Departmental Investment Funds
(Earth & Life Sciences), Natural History Museum, London. TR is funded by a
University of Brighton Science Scholarship.

**References**

1. Alvarez LW, Alvarez W, Asaro F, Michel HV. 1980 Extraterrestrial cause for the
cretaceous-tertiary extinction. *Science* **208**, 1095–1108.
2. Schulte P *et al.* 2010 The Chicxulub asteroid impact and mass extinction at the
Cretaceous-Paleogene boundary. *Science* **327**, 1214–1218.
3. Brusatte SL *et al.* 2015 The extinction of the dinosaurs. *Biol. Rev. Camb. Philos. Soc.*
**90**, 628–642.
4. Chiarenza AA, Farnsworth A, Mannion PD, Lunt DJ, Valdes PJ, Morgan JV, Allison PA.
2020 Asteroid impact, not volcanism, caused the end-Cretaceous dinosaur extinction.
*Proc. Natl. Acad. Sci. U. S. A.* (doi:10.1073/pnas.2006087117)
5. Kaiho K, Oshima N, Adachi K, Adachi Y, Mizukami T, Fujibayashi M, Saito R. 2016
Global climate change driven by soot at the K-Pg boundary as the cause of the mass
extinction. *Sci. Rep.* **6**, 28427.
6. Gulick SPS, Morgan JV, Scientists I-IE 364. 2019 What has Chicxulub Taught Us About
Large Impact Processes and Mass Extinction? *LPI Contributions* **2136**, 5114.
7. Artemieva N, Morgan J. 2020 Global K-Pg Layer Deposited From a Dust Cloud.
*Geophys. Res. Lett.* **47**, 289.
8. Wang SC, Dodson P. 2006 Estimating the diversity of dinosaurs. *Proc. Natl. Acad. Sci.*
*U. S. A.* **103**, 13601–13605.
9. Upchurch P, Mannion PD, Benson RBJ. 2011 Geological and anthropogenic controls
on the sampling of the terrestrial fossil record: a case study from the Dinosauria.
*Geological Society*
10. Chiarenza AA, Mannion PD, Lunt DJ, Farnsworth A, Jones LA, Kelland S-J, Allison PA.
2019 Ecological niche modelling does not support climatically-driven dinosaur diversity
decline before the Cretaceous/Paleogene mass extinction. *Nat. Commun.* **10**, 1091.
11. Sakamoto M, Benton MJ, Venditti C. 2016 Dinosaurs in decline tens of millions of years
before their final extinction. *Proc. Natl. Acad. Sci. U. S. A.* **113**, 5036–5040.
12. Chenet A-L, Courtillot V, Fluteau F, Gérard M, Quidelleur X, Khadri SFR, Subbarao KV,
Thordarson T. 2009 Determination of rapid Deccan eruptions across the Cretaceous-
Tertiary boundary using paleomagnetic secular variation: 2. Constraints from analysis of
eight new sections and synthesis for a 3500-m-thick composite section. *J. Geophys.*
*Res. [Solid Earth]* **114**.
13. Huber BT, Norris RD, MacLeod KG. 2002 Deep-sea paleotemperature record of

extreme warmth during the Cretaceous. *Geology* **30**, 123–126.
14. Wild M, Calanca P, Scherrer SC, Ohmura A. 2003 Effects of polar ice sheets on global
sea level in high-resolution greenhouse scenarios. *J. Geophys. Res. D: Atmos.* **108**.
15. Miller KG *et al.* 2005 The Phanerozoic record of global sea-level change. *Science* **310**,
1293–1298.
16. Grossman EL. 2012 Applying oxygen isotope paleothermometry in deep time. *The*
*Paleontological Society Papers*
17. Tobin TS, Ward PD, Steig EJ, Olivero EB, Hilburn IA, Mitchell RN, Diamond MR, Raub
TD, Kirschvink JL. 2012 Extinction patterns, $\delta^{18}\text{O}$ trends, and magnetostratigraphy
from a southern high-latitude Cretaceous–Paleogene section: Links with Deccan
volcanism. *Palaeogeogr. Palaeoclimatol. Palaeoecol.* **350–352**, 180–188.
18. Signor PW, Lipps JH. 1982 Sampling bias, gradual extinction patterns and catastrophes
in the fossil record. *Geological Society of America Special Papers* **190**.
19. Smith AB. 2001 Large-scale heterogeneity of the fossil record: implications for
Phanerozoic biodiversity studies. *Philos. Trans. R. Soc. Lond. B Biol. Sci.* **356**, 351–
367.
20. Benton MJ, Dunhill AM, Lloyd GT. 2011 Assessing the quality of the fossil record:
insights from vertebrates. *Geological Society*
21. Vilhena DA, Smith AB. 2013 Spatial bias in the marine fossil record. *PLoS One* **8**,
e74470.
22. Walker FM, Dunhill AM, Benton MJ. 2020 Variable preservation potential and richness
in the fossil record of vertebrates. *Palaeontology* **63**, 313–329.
23. Dean CD, Chiarenza AA, Maidment SCR. 2020 Formation binning: a new method for
increased temporal resolution in regional studies, applied to the Late Cretaceous
dinosaur fossil record of North America. *Palaeontology* **26**, 23.
24. Barrett PM, McGowan AJ, Page V. 2009 Dinosaur diversity and the rock record. *Proc.*
*Biol. Sci.* **276**, 2667–2674.
25. Lloyd GT. 2012 A refined modelling approach to assess the influence of sampling on
palaeobiodiversity curves: new support for declining Cretaceous dinosaur richness.
*Biol. Lett.* **8**, 123–126.
26. Starrfelt J, Liow LH. 2016 How many dinosaur species were there? Fossil bias and true
richness estimated using a Poisson sampling model. *Philos. Trans. R. Soc. Lond. B*
*Biol. Sci.* **371**, 20150219.
27. Dunhill AM, Hannisdal B, Brocklehurst N, Benton MJ. 2018 On formation-based
sampling proxies and why they should not be used to correct the fossil record.
*Palaeontology* **61**, 119–132.
28. Close RA, Evers SW, Alroy J, Butler RJ. 2018 How should we estimate diversity in the
fossil record? Testing richness estimators using sampling-standardised discovery
curves. *Methods Ecol. Evol.* **9**, 1386–1400.
29. Alroy J. 2020 On four measures of taxonomic richness. *Paleobiology* **46**, 158–175.

30. Benson RBJ, Campione NE, Carrano MT, Mannion PD, Sullivan C, Upchurch P, Evans
DC. 2014 Rates of dinosaur body mass evolution indicate 170 million years of
sustained ecological innovation on the avian stem lineage. *PLoS Biol.* **12**, e1001853.
- 31. Lloyd GT, Davis KE, Pisani D, Tarver JE, Ruta M, Sakamoto M, Hone DWE, Jennings
R, Benton MJ. 2008 Dinosaurs and the Cretaceous Terrestrial Revolution. *Proc. Biol.*
*Sci.* **275**, 2483–2490.
- 32. Cau A, Brougham T, Naish D. 2015 The phylogenetic affinities of the bizarre Late
Cretaceous Romanian theropod Balaur bondoc (Dinosauria, Maniraptora):
dromaeosaurid or flightless bird? *PeerJ* **3**, e1032.
- 33. Carballido JL, Pol D, Otero A, Cerda IA, Salgado L, Garrido AC, Ramezani J, Cúneo
NR, Krause JM. 2017 A new giant titanosaur sheds light on body mass evolution
among sauropod dinosaurs. *Proc. Biol. Sci.* **284**. (doi:10.1098/rspb.2017.1219)
- 34. Raven TJ, Maidment SCR. 2017 A new phylogeny of Stegosauria (Dinosauria,
Ornithischia). *Palaeontology* **60**, 401–408.
- 35. Thompson RS, Parish JC, Maidment SCR, Barrett PM. 2012 Phylogeny of the
ankylosaurian dinosaurs (Ornithischia: Thyreophora). *J. Syst. Palaeontol.* **10**, 301–312.
- 36. Arbour VM, Zanno LE, Gates T. 2016 Ankylosaurian dinosaur palaeoenvironmental
associations were influenced by extirpation, sea-level fluctuation, and geodispersal.
*Palaeogeogr. Palaeoclimatol. Palaeoecol.* **449**, 289–299.
- 37. Mallon JC, Ott CJ, Larson PL, Iuliano EM, Evans DC. 2016 *Spiclypeus shipporum* gen.
et sp. nov., a Boldly Audacious New Chasmosaurine Ceratopsid (Dinosauria:
Ornithischia) from the Judith River Formation (Upper Cretaceous: Campanian) of
Montana, USA. *PLoS One* **11**, e0154218.
- 38. Cruzado-Caballero P, Gasca JM, Filippi LS, Cerda IA, Garrido AC. 2019 A new
ornithopod dinosaur from the Santonian of Northern Patagonia (Rincón de los Sauces,
Argentina). *Cretaceous Res.* **98**, 211–229.
- 39. González Riga BJ, Mannion PD, Poropat SF, Ortiz David LD, Coria JP. 2018 Osteology
of the Late Cretaceous Argentinean sauropod dinosaur *Mendozasaurus neguyelap*:
implications for basal titanosaur relationships. *Zool. J. Linn. Soc.* **184**, 136–181.
- 40. Goloboff PA, Farris JS, Nixon KC. 2008 TNT, a free program for phylogenetic analysis.
*Cladistics* **24**, 774–786.
- 41. Bapst DW. 2012 paleotree : an R package for paleontological and phylogenetic
analyses of evolution : Analyses of Paleo-Trees in R. *Methods Ecol. Evol.* **3**, 803–807.
- 42. Laurin M. 2004 The evolution of body size, Cope's rule and the origin of amniotes. *Syst.*
*Biol.* **53**, 594–622.
- 43. Paradis E, Schliep K. 2019 ape 5.0: an environment for modern phylogenetics and
evolutionary analyses in R. *Bioinformatics* **35**, 526–528.
- 44. Kembel SW, Cowan PD, Helmus MR, Cornwell WK, Morlon H, Ackerly DD, Blomberg
SP, Webb CO. 2010 Picante: R tools for integrating phylogenies and ecology.
*Bioinformatics* **26**, 1463–1464.
- 45. Hadfield JD. 2010 MCMC methods for multi-response generalized linear mixed models:
the MCMCglmm R package. *J. Stat. Softw.*

46. Plummer M, Best N, Cowles K, Vines K. 2006 CODA: convergence diagnosis and
output analysis for MCMC. *R News* **6**, 7–11.
47. R Development Core Team. 2019 R: A language and environment for statistical
computing.
48. Bonsor JA, Barrett PM, Raven TJ, Cooper N. 2020 Dinosaur evolutionary rates were
not in decline prior to the K-Pg boundary. (doi:10.5519/0034257)
49. Chiba K, Ryan MJ, Fanti F, Loewen MA, Evans DC. 2018 New material and systematic
re-evaluation of *Medusaceratops lokii* (Dinosauria, Ceratopsidae) from the Judith River
Formation (Campanian, Montana). *J. Paleontol.* **92**, 272–288.
50. Brown CM, Evans DC, Ryan MJ. 2013 New data on the diversity and abundance of
small-bodied ornithopods (Dinosauria, Ornithischia) from the Belly River Group
(Campanian) of Alberta. *J. Vert. Paleontol.*
51. Brusatte SL. 2012 *Dinosaur Paleobiology*. John Wiley & Sons.
52. Harvey PH, Pagel MD, Others. 1991 *The comparative method in evolutionary biology*.
Oxford university press Oxford.
53. Smith AB. 1994 Rooting molecular trees: problems and strategies. *Biol. J. Linn. Soc.*
*Lond.* **51**, 279–292.
54. Crouch NMA. 2020 Extinction rates of non-avian dinosaur species are uncorrelated with
the rate of evolution of phylogenetically informative characters. *Biol. Lett.* **16**, 20200231.
55. McGowan A, Smith AB. 2011 *Comparing the Geological and Fossil Records:*
*Implications for Biodiversity Studies*. Geological Society of London.
56. Mannion PD, Benson RBJ, Butler RJ. 2013 Vertebrate palaeobiodiversity patterns and
the impact of sampling bias. *Palaeogeogr. Palaeoclimatol. Palaeoecol.*
57. Chenet A-L, Fluteau F, Courtillot V, Gérard M, Subbarao KV. 2008 Determination of
rapid Deccan eruptions across the Cretaceous-Tertiary boundary using paleomagnetic
secular variation: Results from a 1200-m-thick section in the Mahabaleshwar
escarpment. *J. Geophys. Res. [Solid Earth]* **113**.
58. Cooper, N. 2020. GitHub: nhcooper123/dino-trees: Code for the paper. DOI:
10.5281/zenodo.3935932.

Appendix B

Response to reviewers

We thank the reviewers and editor for their helpful comments. We have pasted these below, with our responses following each in red text.

We fully appreciate the comments below and understand the logic and desire to provide some sensitivity analyses related to how the trees were time calibrated and presented (Reviewer 2). Unfortunately, what appear to be simple changes are not simple in both the context of the paper and our current situation in terms of home-working during the current pandemic. The full set of MCMCglmm models takes almost one month to run if we include the models with intercepts set to 1 (see Reviewer 1). Therefore, each sensitivity analysis suggested below adds (at least) a month of computational time to our revision (and we currently lack access to a cluster due to home connectivity issues, so the analyses are all being run on a laptop that is also being used to work on other things simultaneously). We have therefore done our best to run as many analyses as possible within the revision time frame (originally one week, extended to 2 weeks), and hope these are sufficient to allay the concerns of the reviewers without committing to several months of running non-stop analyses.

Reviewer: 1

I support the publication of this work with minor revision. It presents quite an important set of analyses that probe the findings of Sakamoto et al (2016, PNAS) and add considerably to our ability to interpret those results. This is important because it relates to a long-standing debate about dinosaur decline in advance of the K-Pg extinction event. It also has importance in the development of phylogeny-based approaches to the study of diversification in the fossil record — this area has longer-term promise to unravel some key questions about macroevolutionary diversification in general. The paper has potential to be a classic in this area as it provides one of the most critical and reflective applications of these methods so far.

I wholeheartedly endorse publication in Biology Letters. It is certainly of appropriate significance and I would be disappointed if it was not published.

We thank the reviewer for these supportive comments.

My key suggestions are as follows:

(1) The intercept of the models could be constrained to fall between 0 and 1.0, or set to 1.0 instead of 0. I believe that 1.0 nodes (i.e. the root node) should be present at even infinitesimally small values of time. Also, DIC could be used to evaluate models with constrained intercepts compared to those with estimated intercepts.

Constraining the intercepts within the GLMM framework isn't easy to implement, so we have instead repeated the analyses setting the intercepts to 1.0. We expand on this in the comments to specific points below.

(2) Various minor suggestions about wording. Plus mentioning issues of sampling rate variation and potential solutions (even if not implemented here) in the introduction. They form an important theme of the discussion after all.

Agreed. See comments to specific points below.

(3) Some relevant literature is not cited - you know, it's literature from my group so you can do eye-rolling if needed. I tried to restrict the amount that I suggested this however.

Apologies for the oversight here – we have updated as appropriate.

SPECIFIC COMMENTS

Title: "Evolutionary rates" can mean several different things. It may be more informative instead to write 'diversification rates'.

Changed throughout.

Lines 53-56: "Determining trends in Mesozoic dinosaur species-richness is a contentious issue and has been the focus of numerous previous studies (e.g. [3,8–10]). Many different hypotheses have been proposed, but there are currently two major schools of thought".

Change "Determining trends in Mesozoic dinosaur species-richness" to "Determining trends in Mesozoic dinosaur species-richness *leading up to the K-Pg*". That would reflect the focus of this paragraph more accurately and avoid the misleading implication that studies of dinosaur species richness are restricted to opinions about the K-Pg.

Changed.

Also, I think it would be more instructive to suggest three different viewpoints here, because Sakamoto's study on its own does not reflect a 'major school of thought', whereas an actual 'major school of thought' is missing here. My view of the literature is that two things have been widely suggested: (1) No decline in advance of the boundary. (2) A decline on the timescale of 100,000s or millions of years related to non-bolide factors such as Deccan flood volcanism (older literature reviewed by Alvarez 1983 PNAS, and also see Dean et al 2020, cited in text, for more recent literature). (3) Sakamoto presented a third view that there was a decline on the timescale of 10s of millions of years without noting that no-one really had suggested this previously.

The main text has been updated with these suggestions and clarifications on the theories.

Lines 75-78: "Reconstruction of past diversity patterns is difficult, as the 'raw' patterns of taxonomic diversity gleaned from the fossil record are subject to many hierarchical biases, ranging from initial fossilization potential, through the availability of fossiliferous rock outcrop, to anthropic collection biases (e.g. [18–23])"

Refs 18-23 include various potentially-appropriate citations about fossil record sampling heterogeneity. Some of them are classics (Signor & Lipps 1982 on edge effects). Some of them are relatively unfocussed studies that provide laundry lists of untested qualitative points. Vilhena and Smith (2013) is a study of spatial bias in the marine fossil record that doesn't really propose a solution to the problem.

A reference to Vilhena and Smith (2013) has been added to the above paragraph.

Work presenting new methods to address spatial-bias and applying that to the dinosaur fossil record is not cited.

Close RA, Benson RBJ, Alroy J, Carrano MT, Clearly T, Dunne E, Mannion PD, Uhen M, Butler RJ. 2020. The apparent exponential radiation of Phanerozoic land vertebrates reflects spatial sampling biases. *Proceedings of the Royal Society B* 287, 20200372.

Cited where appropriate.

I also think it might be appropriate to cite Benson (2018, *Dinosaur macroevolution and macroecology*) which contains discussions and plots related to dinosaur diversity dynamics, spatial sampling variation, regional heterogeneity, Sakamoto et al and other topics that are discussed in the current paper. Some of the statements in Section 5 of Benson (2018), on several different topics, are quite similar to those of the current paper.

Reference added to the appropriate section.

Lines 78-81: "Different methods have been applied in attempts to correct for these factors, including various subsampling techniques (rarefaction, Shareholder Quorum Subsampling and TRiPs) and comparisons of diversity with collections-related proxies"

Spatial bias is emphasised in earlier text, but none of the methods written into the text here can correct for spatial bias. New approaches to address spatial bias were presented by Close et al (2020), which also report patterns of dinosaur diversity.

Thank you for bringing this recent paper to our attention, reference added.

Lines 82-83: "...with substantial disagreement over the patterns recovered (e.g. [3,8,9,24–29])".

Refs 27-29 are papers evaluating method performance rather than papers about patterns of dinosaur diversity. So they seem to be cited in the wrong place. Arguably, they don't need to be cited at all unless the text includes statements about method performance.

Inappropriate references removed.

Lines 85-137: These paragraphs list potential problems with the analyses/interpretation of Sakamoto et al. There is another problem that is not mentioned in the text but is potentially very important. Fossil phylogenies are different to phylogenies of living species because times represent sampling events. Sampling events also influence the node counts, because each sampling event adds one tip during the sampled interval and one node during some older interval. Sakamoto et al didn't really do anything that would be appropriate to address this but a lot of progress was made on appropriate models by other groups — primarily Tanja Stadler's group and the fossilise-birth-death (FBD) model. Although it might not be tractable to apply that to this problem right now, I think it would be right to mention this shortcoming and that solutions might be possible in future using models such as the FBD model, even though that is not applied here.

Thanks for bringing this to our attention, although beyond the time constraints of this study, it's certainly something we would consider for any future studies.

Lines 143-145: "two subtly different versions of the Benson et al. [30] supertree, which reflect plausible differences in the relationships of several taxa".

The trees reflect difference in hypotheses of sauropod inter-relationships. This could be stated instead of "relationships of several taxa".

Changed.

Line 146: "first occurrence date (FAD) or last occurrence date (LAD)"

These mostly aren't FADs and LADs. Most are the oldest-age and youngest-age bounds on uncertainty associated with taxon ages. So, the same as "Maximum and minimum possible ages" (lines 160-161).

The FAD and LAD dating was originally referred to by Sakamoto *et al* (2016), and we state in this paragraph how those authors carried out their original analysis, and how their original trees that we then also use in our own analyses were dated.

Lines 190-191: "As closely related species are likely to have more similar node count values" By definition they -must- have similar node counts. If you agree, consider rephrasing 'are more likely to have'.

Changed.

Line 228: "776 (86%) did not favour any model at all"

I understand entirely why this is written. However, the statement doesn't contain much information. I'd really like to see the densities for the Lloyd et al and Benson et al. results but they aren't shown in Figs 2 or 4. There shouldn't be an issue with space-availability to include these because the panels aren't complex and their sizes could be reduced.

We didn't include the density plots for these initially because each tree only has three different dating methods (rather than the 100 for each of the other 9 trees). However, we have now added these as suggested to Figures 2 and 4 (and new Figure S3). Figure 4 is now Figure S1, see later comments.

Figures 2 and 4.

It would be useful to write out the name of the dinosaur subgroup on each panel. It's a bit tricky to figure it out as currently-presented. You can make it very easy for readers to see at-a-glance what groups are analysed.

We have now added the small dinosaur icons to indicate the groups involved in each tree for Figures 2 and 4 (and new Figure S3). Figure 4 is now Figure S1, see later comments.

Lines 296-298: "Their conclusions might also have been skewed by reliance on estimated intercepts [11] rather than zero-intercepts, as the latter would, perhaps, be more biologically plausible (as there should be no new nodes at t0)"

I agree that setting the intercept to zero has a big effect on the result - that's clear from the analyses. There are a few things to say on this. Firstly: I'm not sure how many nodes there should be at $t = 0$, because the count of nodes at $t = 0$ does not have a straightforward interpretation. In reality, at $t > 0$, even at infinitesimally small values, at least 1.0 nodes must have occurred because $t = 0$ is the time of the most basal node in the tree. The fact that infinitesimally

small values are essentially equal to zero is important and suggest that when time = 0, the node count should be 1.0. However, zero tips have occurred at time = 0, so in reality we haven't had any data-items that would record time = 0 and nodes = 1. By definition, zero tips occur until time = 1 (because of the algorithm use to scale the trees to time). By time = 1 all we know is that node counts are greater than or equal to 1.0.

These are my recommendations:

- (1) Set the intercept to 1.0 instead of or as well as 0. Or constrain the value of the intercept to be between 0 and 1.0 but don't specify it precisely.
- (2) Use DIC to evaluate which model is best (i) a model in which the intercept is entirely unconstrained, (ii) a model in which the intercept is between 0 and 1.0, (iii) any other intercept setting that you like.

We have followed these recommendations as follows.

(1) We repeated the analyses with the intercepts set to 1.0. This was preferred over constraining the intercepts to vary between 0-1 as it was computationally simpler to implement. New results are in Figures S3-4 and Table S3 and discussed in the text.

(2) (i-ii) We looked at the DIC values to determine which was the “best” approach with respect to intercept fitting. All favoured either the intercepts estimated models or none of them (i.e. there wasn't a big difference in DIC whether intercepts were estimated or set to zero or one). While this suggests the estimated intercept models fit the best overall, we didn't want to be overly reliant on this because for some trees (e.g. the Benson and Lloyd trees), the estimated intercepts are a bit larger than we'd theoretically predict they should be (close to 2.5), so although the model fit is better, the biological fit is poorer. However, we take the point that our protocol is a little confusing, so have reduced emphasis on the differences in intercepts between our paper and the Sakamoto paper, and have relegated the figures for the zero and one intercepts results to the supplementary materials, although we still describe the results in the text.

(2)(iii). We did not use any other intercepts due to the computational time taken to run the analyses just with one different intercept (~1 week), and because setting the intercept to 1.0 didn't produce very different results to setting the intercept to zero (see Tables S2 and S3, and Figs S1-S4).

We added to the methods:

“Defining the theoretical value for the node count when no time has elapsed is not straightforward. The trees have no tips at t_0 , meaning that node count will be zero, however, because the tree has a root node, there is technically a node count of 1. We therefore fitted models where the intercept is estimated (following Sakamoto et al [16]), with the intercept set to zero, and with the intercept set to 1.0, to determine how this influenced our results.”

We added to the results:

For models where intercepts were set to 1.0 (Figs S3 and S4; Table S3), 183 (20.3%) of our 900 new trees unambiguously favoured the downturn model, 209 (23.2%) unambiguously favoured the slowdown to asymptote model, and 469 (52.1%) favoured either the downturn or the slowdown to asymptote model. No trees favoured the null model and 39 (4.33%)

favoured no model at all (Fig. S3; Table S3). All three Lloyd et al. [34] supertrees unambiguously favoured the slowdown to asymptote model, as did three of the Benson et al. [33] supertrees. One Benson et al. [33] supertree unambiguously favoured the downturn model and the remaining two Benson et al. [33] supertrees favoured either the downturn or the slowdown to asymptote model (Fig. S4; Table S3).

We modified the Discussion (see response to comment below) to remove speculation about the role of intercepts in our model interpretations and instead just mention that the best model choices were sensitive to how we fitted the intercepts.

“Our conclusions also appear to be sensitive to the choice of, or estimation of, intercepts in the models.”

Lines 297-298: "would, perhaps, be more biologically plausible"

"perhaps" — it's very British to write this way. But it's also quite vague. This isn't a matter of 'biological plausibility'. In fact, it is axiomatic in the analysis. So I think you should include much stronger statements about this.

This sentence has now been removed - see comments below.

The methodological decisions would also be clearer to me if the estimated intercept values were reported in a graph or similar. If the argument is about 'biological plausibility', so this is quite important. Some values of the intercept are presumably more plausible than others (e.g. a value of 1.0 is quite plausible, see above).

We have now provided an additional graph (Figure S5) demonstrating the posterior means for the intercepts for each of these models. The intercept values don't vary too much - they go up to 2.5 and down to zero (although the posterior distributions they were drawn from are wider obviously), but are rarely exactly 1.0. But we take the reviewers point that all of these values could be argued to be biologically plausible. Given this, allowing the intercept to vary may not be such a big issue for the interpretation of these models. We have therefore removed speculation about this from the discussion, and instead highlight the sensitivity of the “best” model choices in relation to the intercepts as follows:

“Our conclusions also appear to be sensitive to the choice of, or estimation of, intercepts in the models.”

Lines 311: "support the idea that dinosaurs were in terminal decline:

Change to "support the hypothesis that dinosaurs were in terminal decline"

Changed.

Lines 334-337: "Our results, and those of many others (e.g. [3,9,10,23,50]), emphasize the fact that the dinosaur fossil record suffers from numerous gaps and biases, and that any apparent decline in diversity could be due to systemic sampling errors"

This may be the view of the authors (and is also my view - I agree with you). However, it is not supported by any of the analyses presented here, which don't contain any test of the effect of sampling rate variation. See also my note on the Introduction (above) - lines 85-137 do not cite

sampling rate variation as a potential issue with this type of analysis. However, that does seem to be the view of the current authors.

References removed and the statement left as our own opinion based on our results.

Lines 351-356: "For example, a recent study found no correlation between rates of dinosaur morphological evolution and extinction, which suggests that analyses based on existing phylogenetic datasets might not be useful for addressing the question of non-avian dinosaur extinction [54]"

It isn't clear why that result would lead to the conclusion stated here unless we assumed a priori that diversification rates must be correlated with phenotypic rates. Even if we did assume that, then the result might imply that the phenotypic rates are inaccurate (likely, given what they are based on).

Added the text *"however, this result may also indicate inaccuracies in the calculation of phenotypic diversification rates"* to the section detailed above.

Reviewer: 2

The manuscript of Bensor and colleagues seek to address on the main questions regarding the extinction of non-avian dinosaurs, were evolutionary rates decelerating or very low before their extinction? Their main goal is to test the results of a recently published paper that found a downturn of evolutionary rates for most dinosaur clades close to the end of the Cretaceous Period (Sakamoto et al., 2016). I consider that the manuscript is very well written, the figures are very informative, and the conclusions built on well supported results. I haven't found major issues in the manuscript and I have only some moderate comments, but they may imply further analyses or to re-run some analyses (see below). I also made some minor comments and suggestions in an edited version of the manuscript. As a result, I think that the manuscript requires a minor to moderate revision before acceptance.

We thank the reviewer for their positive comments on the MS.

I understand that you haven't used the stochastic dating method 'cal3'. Can you briefly explain why you haven't used it and you prefer a partially stochastic dating but setting a fixed minimum branch length?

If we had been starting from scratch, we would have used cal3. However, we were starting with the trees from Sakamoto et al. that were dated using the partially stochastic method outlined in the manuscript. To be consistent with this, as our intention was to replicate and test their conclusions, we chose to use the same method to date our additional trees. We have added a note to the supplemental methods to clarify this, and added the text:

"We chose this protocol for consistency with the dated trees in Sakamoto et al [2016]."

You have used a minimum branch length of 1 million years, which it is OK for most datasets. However, if you have large trees, it may result in artificially very old root ages. Can you report the mean and standard deviation of the root ages of each of the eighteen trees (9 new trees and 9 trees from Sakamoto et al. 2016, isn't?). After doing that, please, evaluate if the root ages are compatible with the inferred origin of the clades (e.g. that the origin of Theropoda is not occurring

in the Permian). If the latter is the case for any of the trees/clades, I suggest using a shorter minimum branch length or fixing the root of the tree using the inferred minimum divergence time between clades.

Note that we did not date the trees used in Sakamoto et al. We use the dated trees from that paper as the aim of the study was to replicate their analyses. We did however have to date the other nine trees, resulting in 900 time calibrated trees (100 for each).

As suggested, we extracted the root age for each of the 900 trees dated in this study. We present the results as Figure S6 and reproduce it below. The dotted black line shows the oldest inferred origin date for the clade, i.e. 201.3 Ma (Early Jurassic) for Ornithischia and Sauropodomorpha, and 231.4 Ma (Carnian) for Theropoda. The red and blue dashed lines show the oldest and youngest ages respectively of taxa in the trees (dates taken from the Paleobiology Database). Note that the majority of the trees have younger (rather than older as feared) root ages, with the exception of the Raven & Maidment (2017) ornithischian tree, and Carballido et al. (2017) sauropod tree. In these cases, the trees contained outgroup taxa that are older than the group, and the dates instead fall within the oldest and youngest ages of the taxa in the trees. This suggests that our dating method is not biasing trees to be too old and thus the minimum branch length of 1 million years is appropriate for the analyses we present here.

We have added a version of this text, along with the figure below, to the supplementary materials. We also refer readers to this material in the text where we describe our time scaling protocols.

Figure S6: Root ages for each of 900 trees in this study. The dotted black line shows the oldest inferred origin date for the clade, i.e. 201.3 Ma (Early Jurassic) for Ornithischia and Sauropodomorpha, and 231.4 Ma (Carnian) for Theropoda. The red and blue dashed lines show the oldest and youngest ages respectively of taxa in the trees. Panels are ordered based on the dinosaur group in each tree as follows. Ornithischia: Arbour et al. 2016, Chiba et al. 2018, Cruzado-Caballero et al. 2017, Mallon et al. 2016; Raven & Maidment 2017, Thompson et al. 2012; Sauropodomorpha: Carballido et al. 2017, Gonzàlez Riga et al. 2018; Theropoda: Cau et al. 2015.

Different minimum branch lengths may have strong impacts in the results. Thus, I would like to see how sensitive are the results to calibrations using at least two other minimum branch lengths, e.g. 0.1 myr and 5 myr.

Given our findings in response to the comment above, we feel this is not necessary and should not impact the results. We would prefer to avoid running these analyses if possible due to computational limitations associated with working from home, and also feel these will not alter our overall conclusions.

In the supplementary information you state that you have used a 50% majority rule consensus tree for some analyses (e.g. Arbour et al. 2016). There is no reason to exclude some topologies only because they occur in less than 50% of the most parsimonious trees if you are running a cladistic analysis. So, use a strict consensus tree or a strict reduced consensus tree (with wildcard taxa pruned a posteriori) instead.

We made these methodological choices based on the papers the trees were taken from. We wanted to use the trees that the authors of these studies presented as the most strongly supported, following the expert opinions of these authors on their study taxa. We have added a note in the supplemental materials to clarify, and added the following sentence to the main text:

“We used the same methods used in these papers to infer the trees, as our aim was to recreate these published trees, not to provide new phylogenetic inferences.”

Some of these taxon-character matrices are relatively large (>70 taxa). In some of them you have used the New Technology algorithms of TNT during the tree searches. However, 100 replicates of these algorithms may not generate a good sample of optimal results (e.g. Cau analysis). A more appropriate protocol is to use this combination of algorithms but until an a priori determined number of optimal results is reached. You can do this with the TNT command ‘xmult = hits 100;’. I understand that you have followed the search protocols of the original papers, but you should be sure that the tree searches are properly done. It is the same for the consensus trees (see above). I suggest doing searches using the New Technology algorithms if you have more than 70 taxa.

We made these methodological choices based on the papers the trees came from. We weren't interested in inferring the best tree for each dataset per se, instead we wanted to recreate the trees presented in the published papers that the authors felt best represented the clades that they are experts on. Perhaps some of these analyses could have been improved if our primary interest was in inferring the trees. But this was not our primary aim. We have added a note in the supplemental materials to clarify, and added the following sentence to the main text:

“We used the same methods used in these papers to infer the trees, as our aim was to recreate these published trees, not to provide new phylogenetic inferences.”

I can't see a good reason to exclude taxa before the time calibrations. Can you explain why you are doing that in a few analyses?

As we were not attempting to provide novel phylogenetic treatments for any of the clades concerned, we followed the methodologies used by the original authors of the published matrices. We matched any a priori or a posteriori pruning of taxa and used the same tree searching methodologies and settings from the papers the trees were originally published in. Our aim was to recreate the trees published by the taxon-experts for each group. We have added a note in the supplemental materials to clarify, and added the following sentence to the main text:

“We used the same methods used in these papers to infer the trees, as our aim was to recreate these published trees, not to provide new phylogenetic inferences.”

Comments from PDF from Reviewer 2:

Line 45: “Late” removed and replaced with “late”

Line 156 - "Check": Search settings checked and verified

Line 172 - "Please, can you explain here that you have 900 new trees and nine from Sakamoto et al.?" : This is explained in the last sentence of the paragraph above

Line 193 - "one of the triangles below/above the diagonal or the entire matrix?": We weren't sure of the exact meaning of this comment, but we have checked each figure carefully for readability and clarity and corrected where appropriate whilst addressing previous comments from both reviewers.

Line 253 (Figure 2) - "It is not very clear. Can you add a label on the graphic or a silhouette representing the clade?":

We have now added clade silhouettes to Figures 2 and 4 (now Figures 2 and S1) to match those in Figures 3 and 5 (now Figures 3 and S2). And also to new Figure S3.